Biochar’s role in improving pakchoi quality and microbial community structure in rhizosphere soil

Wu Xia 1 2 3
Yang Fengjun yangfengjun@byau.edu.cn yangfengjun@126.com 1
Zhang Jili 4
Gao Feng 1
Hu Yi Chen 1
Yang Kejun 2 4
Wang Peng wangp.ycs@163.com 2 4
1 College of Horticulture and Landscape, Heilongjiang Bayi Agricultural University , Daqing , Heilongjiang , China
2 Post-doctoral Workstation of Agricultural Products Processing Quality Supervision, Inspection and Testing Center (Daqing), Ministry of Agriculture , Daqing , Heilongjiang , China
3 Heilongjiang Bayi Agricultural University, Ministry of Agriculture and Rural Aûairs, Key Laboratory of Low-carbon Green Agriculture Carbon in Northeastrn China , Daqing , Heilongjiang , China
4 College of Agriculture, Heilongjiang Bayi Agricultural University , Daqing , Heilongjiang , China
Sayyed R.Z.
Electronic publication date: 2024 Mar 18
Publication date: 2024
Volume: 12
Electronic Location ID: e16733
Received 2023 May 1; Accepted 2023 Dec 7
Copyright: ©2024 Wu et al.
Copyright year: 2024
Copyright holder: Wu et al.
License: This is an open access article distributed under the terms of the Creative Commons Attribution License, which permits unrestricted use, distribution, reproduction and adaptation in any medium and for any purpose provided that it is properly attributed. For attribution, the original author(s), title, publication source (PeerJ) and either DOI or URL of the article must be cited.
License URL: https://creativecommons.org/licenses/by/4.0/

Keywords: Biochar, Pakchoi, Soil enzymes, Nitrate accumulation, Microbial diversity

Funding: National Natural Science Foundation of China 31801905 Natural Science Foundation of Heilongjiang Province, China LH2021C066 Heilongjiang Bayi Agricultural University Research Program, China XDB201819 This study was jointly funded by the National Natural Science Foundation of China (funding code: 31801905), the Natural Science Foundation of Heilongjiang Province, China (funding code: LH2021C066), and the Heilongjiang Bayi Agricultural University Research Program, China (funding code: XDB201819). The funders had no role in study design, data collection and analysis, decision to publish, or preparation of the manuscript.

==============================
Background

Biochar amendments enhance crop productivity and improve agricultural quality. To date, studies on the correlation between different amounts of biochar in pakchoi (Brassica campestris L.) quality and rhizosphere soil microorganisms are limited, especially in weakly alkaline soils. The experiment was set up to explore the effect of different concentrations of biochar on vegetable quality and the correlation between the index of quality and soil bacterial community structure changes.

Methods

The soil was treated in the following ways via pot culture: the blank control (CK) without biochar added and with biochar at different concentrations of 1% (T1), 3% (T2), 5% (T3), and 7% (T4). Here, we investigatedthe synergistic effect of biochar on the growth and quality of pakchoi, soil enzymatic activities, and soil nutrients. Microbial communities from pakchoi rhizosphere soil were analyzed by Illumina MiSeq.

Results

The results revealed that adding 3% biochar significantly increased plant height, root length, and dry weight of pakchoi and increased the contents of soluble sugars, soluble proteins, Vitamin C (VC), cellulose, and reduced nitrate content in pakchoi leaves. Meanwhile, soil enzyme activities and available nutrient content in rhizosphere soil increased. This study demonstrated that the the microbial community structure of bacteria in pakchoi rhizosphere soil was changed by applying more than 3% biochar. Among the relatively abundant dominant phyla, Gemmatimonadetes, Anaerolineae, Deltaproteobacteria and Verrucomicrobiae were reduced, and Alphaproteobacteria, Gammaproteobacteria, Bacteroidia, and Acidimicrobiia relative abundance increased. Furthermore, adding 3% biochar reduced the relative abundance of Gemmatimonas and increased the relative abundances of Ilumatobacter, Luteolibacter, Lysobacter, Arthrobacter, and Mesorhizobium. The nitrate content was positively correlated with the abundance of Gemmatimonadetes, and the nitrate content was significantly negatively correlated with the relative abundance of Ilumatobacter. Carbohydrate transport and metabolism in the rhizosphere soil of pakchoi decreased, and lipid transport and metabolism increased after biochar application.

Conclusion

Overall, our results indicated that applying biochar improved soil physicochemical states and plant nutrient absorption, and affected the abundance of dominant bacterial groups (e.g., Gemmatimonadetes and Ilumatobacter), these were the main factors to increase pakchoi growth and promote quality of pakchoi. Therefore, considering the growth, quality of pakchoi, and soil environment, the effect of using 3% biochar is better.

Introduction

Biochar is a highly aromatic and refractory solid substance produced by the pyrolysis and carbonization of plant biomass under complete or partial hypoxias (Sun et al., 2016). Due to its porous structure, high surface area, and strong adsorption capacity, it is commonly used as a soil amendment for sustainable agricultural purposes. Numerous scholars have confirmed that adding biochar can improve the physiological characteristics and growth of vegetables, and can increase yield and quality under adverse conditions. Research has shown that the application of biochar and wood vinegar alone or together can improve the yield and nutritional quality of blueberries (Zhang et al., 2020b). Research has displayed that biochar addition can significantly enhance tomato growth, yield, and quality under reduced nitrogen application (Guo et al., 2021). The use of biochar and chitosan in heavy metal contaminated soil has improved the yield and quality of cultivated eggplants (Turan et al., 2018). The combined use of biochar and arbuscular mycorrhizal fungi has a significant impact on the root morphology and chlorophyll content of Fenugreek (Jabborova et al., 2021a). Biochar can improve the performance of the rhizome of ginger and increase the activity of soil enzymes, thereby improving soil nutrient supply (Jabborova et al., 2021b). The effect of biochar on improving fruit quality is related to the improvement of acidic soil properties (Wu et al., 2020; Huang et al., 2022a; Huang et al., 2022b). The impact of biochar on vegetable quality in weakly alkaline soil deserves attention.

Applying biochar can change soil’s physical and chemical properties and improve the growth and quality of crops. Soil improvement with biochar can enhance soil stability and fertility because biochar enhances soil properties and nutrient availability (You et al., 2021). Soil-applied biochar can mitigate nitrate leaching (Llovet et al., 2021) and significantly increase the available levels of nitrogen (N) in the soil (Li, Liang & Shangguan, 2017). Additionally, the individual or co-application of biochar and wood vinegar increases soil nutrient availability (e.g., NH4+-N, NO3−-N, and Mg2+) (Zhang et al., 2020b). Borchard et al. (2012) found that the physical activation of biochar decreased the contents of NO3−-N and P by approximately 55% and 90% (w/w), respectively. Different biochar samples, except for N-rich biochar, exhibited minor N release after successive batch extractions. Ammonium was the primary form of N released from biochar, followed by organic N. Nitrate was in the range of 2%–30% in the leachates, whereas organic N was up to 59%. The release of dissolved organic carbon, N, and P into the soil solution was significantly correlated with biochar volatile matter content and acid functional group density (Mukherjee & Zimmerman, 2013). Adding biochar improved nitrogen utilization by adsorption and release of nitrogen. However, the correlation between nitrate accumulation in vegetables by biochar and the balance of soil nutrients has become an important issue.

Biochar as a soil amendment can stimulate microbial processes in organic farming, resulting in better vegetable production and improved soil properties for sustainable farming (Jun et al., 2016). Research depicts that the highly porous structure of biochar provides a habitat for microorganisms to settle, allowing them to grow better in the soil environment (Wong & Ogbonnaya, 2021). Research has demonstrated that biochar with organic–inorganic fertilizer improves soil nutrients, enzymatic activities, and bacterial abundance (Song et al., 2022). Firmicutes, Proteobacteria, Acidobacteria, Bacteroidetes, and Actinobacteria were the dominant phyla in all soil samples, and applying biochar affected the abundance of dominant bacterial groups (Zheng et al., 2021). Evidence displays that biochar amendment shifts bacterial community structures by altering soil chemical properties (Yu et al., 2012; Zheng et al., 2018). The short-term application of vinasse biochar can change the structure and diversity of the soil bacterial community and control the risk of soil nitrogen leaching by inhibiting ammonia oxidation and nitrification of soil to improve soil nitrogen availability (Zhang et al., 2020a). Biochar application level influences the structure and diversity of the soil microbiome and plant performance (Meng et al., 2018). However, there is a gap in the study of the effect of biochar on nitrate content in vegetables and the interactions between soil microbial communities. Biochar can bioremediate organic contaminants by harboring microbial populations, releasing contaminant-degrading enzymes and protecting beneficial microorganisms from the immediate toxicity of surrounding contaminants (Mukherjee et al., 2022; Zhou et al., 2022b).

The biological mechanism by which biochar improves the quality of pakchoi remains unclear. We aimed to test the following three hypotheses: (1) biochar can improve growth and quality of pakchoi; (2) biochar can enhance soil enzymatic activities and soil available nutrient content; (3) biochar can improve the quality of pakchoi by altering the abundance of soil dominant bacterial communities and altering the soil nutrient environment. The purpose of this study is to evaluate the improvement effect of different concentrations of biochar on the rhizosphere soil of pakchoi, and to analyze the correlation between biochar and soil microbial and quality improvement. To this end, physiological and biochemical methods were used to estimate soil physicochemical properties and tomato nutrient uptake, and Illumina MiSeq sequencing was used to analyze the abundance and composition of soil microbial communities.

Materials and Methods

Experimental materials

In this study, the soil was obtained from the experimental base of the Agronomy College of Heilongjiang Bayi Agricultural University. The soil is weakly alkaline chernozem with the following physical and chemical properties: contained 156 mg kg-1 alkali-hydrolyzable nitrogen, 14.99 mg kg-1 phosphorus, 100 mg kg-1 potassium, and 0.478 mg kg-1 heavy metal cadmium, with 3.78% organic matter, and the soil’s pH was 8.3. This research acquired the pakchoi variety of “ Siji Xiaobaicai” (Year-round plantable pakchoi). The biochar was prepared with corn stalk (preparation temperature was 450 °C, provided by Anhui Xiangzheng Biological Engineering Co., Ltd., pH 8.41, EC 1.07 mS cm−1, NH4+-N 7.61 mg kg−1, NO3−-N 30.82 mg kg−1, available P 97.12 mg kg−1, available K 3,012.00 mg kg−1, bulk density 0.35 g cm−3.

Experimental design

The experiment was conducted in the Heilongjiang Bayi Agricultural University greenhouse from May to June 2019. CK1, without biochar, was used as a control. Four treatments (Table 1) with 1% (T1), 3% (T2), 5% (T3), and 7% (T4) biochar were repeated thrice in the test facility. The cultivated soils were filled into a plastic basin with a 22 cm height and an 18 cm diameter. The same compound fertilizer and chicken manure were added to each pot. The sowing and thinning management were kept constant for all experiments. Samples were collected after 30 days (seedling stage) of growth to determine the relevant indexes of pakchoi.

Experimental methods

When pakchoi samples were collected after 30 days (seedling stage) of growth, plant height and root length were measured using a meter ruler. The aboveground and underground parts of pakchoi were rinsed, leaf moisture was dried using filter paper, and the plant was weighed on a scale to obtain the fresh weight. The plants were wrapped in paper, placed in an oven at 105 °C for 30 min to remove water, and dried at 75 °C to obtain dry weight.

This study took 0.1 g from the fifth leaf of fresh pakchoi in a leaching solution (alcohol: acetone = 1:1) for 24 h, and colorimetry was performed at 663 and 645 nm to calculate the contents of chlorophyll a, chlorophyll b, and the overall amount of chlorophylls (Zhang, 1992). A 10 g sample of pakchoi and 5 ml of 2% oxalic acid solution were ground in a mortar. The ground sample was diluted to a 100 ml volumetric flask with 2% oxalic acid, filter, and a 10 ml of the filtrate was placed in an evaporating dish. It was titrated with a calibrated 2,6-dichlorophenol indophenol solution until it turned pink and did not fade within 30 s. The amount of dye used was recorded and the content of vitamin C was calculated (Li, 2010). The method established by Li (2010) was used to determine the soluble sugar content. A total of 0.2 g of the sample to 20 ml of distilled water was added and extracted in a water bath for 30 minutes. Then, 1.5 ml of distilled water was added to 0.5 ml of the extraction solution. Then, 0.5 ml of ethyl acetate and 5 ml of concentrated sulfuric acid were introduced into the test tube in sequence. After vigorous shaking, the mixture was subjected to a boiling water bath for 1 min. After cooling, absorbance at 630 nm was measured to calculate the soluble sugar content (Li, 2010). A total of 2 g of the sample was ground into a homogenate and extracted in 45 °C water bath for 1 h. A total of 0.2 ml of the extraction solution was added to a test tube containing 5 ml of 5% salicylic acid sulfate solution for 30 min, then 19 ml of NaOH solution was added absorbance was measured at 410 nm, and nitrate content was calculated (Li, 2010). A total of 0.5 g of fresh sample was ground into a homogenate with 5 ml of distilled water, then centrifuged at 10,000 r min−1 for 10 min; then, 1.0 ml of the supernatant was taken and placed in a test tube, 5 ml of Coomassie Brilliant Blue G-250 solution was added, mixed and placed for 2 min at 595 nm for colorimetry. The absorbance was measured, and the protein content was calculated (Li, 2010). A total of 0.2 g of dry sample was placed into a beaker, 100 ml of 60% H2SO4 was added, shaken well, and filtered with a Buchner funnel into another beaker. A total of 2 ml of the filtrate was taken into a stopper tube, 0.5 ml of 2% anthrone reagent was added, and 5 ml of concentrated H2SO4 was added. The stopper was sealed and shaken well, and let to stand for 12 min. The absorbance was measured at a wavelength of 620 nm and the cellulose content was calculated (Xiong, Zuo & Zhu, 2005).

Table 1 Treatments of the experiment.

Treatments	Biochar (%)	Chicken manure (%)	N, P, K Fertilizer (g kg−1 )	
CK	0	2.5	1.18	
T1	1	2.5	1.18	
T2	3	2.5	1.18	
T3	5	2.5	1.18	
T4	7	2.5	1.18	
Notes.

CK, biochar treatment was not applied; T1, treated with 1% biochar; T2, treated with 3% biochar; T3, treated with 5% biochar; T4, treated with 7% biochar. 1, 2, and 3 represent three replications.

For the experimental soil samples in 2019, the soil was sampled by shaking the roots (Wang et al., 2009) after 30 days of growth. For each sampling, the roots from five tomato seedlings were collected and pooled together as one sample with three samples per treatment (n = 3), and sieved using a 20 mesh. Some of these soil samples were stored at 4 °C to evaluate soil enzyme activities. The phenol-sodium hypochlorite colorimetric and disodium phenyl phosphate methods were used to determine soil urease and alkaline phosphatase activities. In addition, 3,5-dinitrosalicylic acid colorimetric, TTC staining, and pyrogallol colorimetric determination of sucrase activity, dehydrogenase activity, and soil polyphenol oxidase activity, respectively, were performed. Soil cellulase activity was determined using 3,5-dinitrosalicylic acid (Zhang & Li, 2005). Another portion of these soil samples was naturally air-dried to determine the alkali-hydrolyzable nitrogen by the alkaline hydrolysis-diffusion-titration procedure in a diffusion disk, the available phosphorus by molybdenum blue colorimetry using a spectrophotometer, and the available potassium by ammonium acetate extraction-flame photometry. The pH and EC values were determined using an acidimeter and a conductivity meter, respectively, after leaching and filtering the soil-water mixture (soil: water = 1:5) (Bao, 2000). Then, 0.5 g of the pulverized dry plant samples were digested with the HNO3-HClO4 (V:V = 4:1) mixed acid system and analyzed using an atomic absorption spectrometer (Emmett, Buckley & Drinkwater, 2020).

For the experimental soil samples in 2019, adopting the MolPure® Soil DNA Kit soil DNA extraction kit (18815ES50) can increase recovery yield by: (1) preheating the eluent at 65 °C; (2) adding the filtrate to the column again, let it sit at room temperature for 3 min, and elute. The sample DNA can be stored at −20 °C for short-term storage and −80 °C for long-term storage. The soil microorganism MiSeq was sequenced by the Shanghai Meiji Biological Company (Shanghai, China). Polymerase chain reaction (PCR) amplification as described by Magoč & Salzberg (2011) then the bacterial v3–v4 and fungal regions were amplified. The primers employed in this investigation were 338f/518r. The PCR products were analyzed using quantifluor™. Each PCR reaction, with a total volume of 20 µL, consisted of 9 µL of 2 × Real SYBR Mixture, 0.2 µL of each primer (10 µM), 2.5 µL of template DNA, and 8.1 µL of ddH2O. The qPCR reaction conditions are as follows: (1) Annealing: Following denaturation, the reaction proceeded to an annealing step. The annealing temperature was set at 65 °C and lasted for 30 s. During this step, the primers bind to their complementary sequences on the DNA template. (2) Extension: After annealing, the reaction underwent an extension step. The extension temperature was set at 72 °C and lasted for 1 min. During this step, the DNA polymerase extends the primers, synthesizing new DNA strands. (3) Final extension: The amplification process was repeated for a total of 22 cycles, followed by a final step of elongation. The temperature was set at 72 °C and lasted for 10 min. This step ensures that any remaining DNA strands are fully extended. (4) Fragment length: The amplified fragment length obtained in this study was approximately 230 base pairs (bp). This length corresponds to the targeted DNA region amplified by the primers. To analyze the quantitative data, a ST Blue Fluorescence Quantitative System (Promega, Madison, WI, USA) was used. Additionally, the MiSeq library was constructed and sequenced to obtain the sequence of DNA fragments (Xu, Wang & Wu, 2015).

Data processing

All collected data were analyzed using IBM SPSS Statistics ver. 18 (SPSS Inc., Chicago, IL, USA). Duncan’s Multiple Range tests (DMRT) and post hoc tests were used to compare the means of the treatments. The significance level was set at P <0.05. The biological information analysis method statistically and visually analyzed the sequencing data to obtain optimized effective sequence statistics, OTU distribution statistics, diversity index analysis, and sample community composition analysis. Finally, a heatmap diagram was plotted (Xu, Wang & Wu, 2015).

Results and Analysis

Effects of different treatments on the growth of pakchoi

The results demonstrated that pakchoi root length, shoot dry weight, and root dry weight significantly increased under 3% biochar treatment (T2) compared to CK. Compared with CK and 3% biochar treatment (T2), the fresh and root dry weight of pakchoi significantly increased under 5% (T3) and 7% biochar treatments (T4). Therefore, the growth of pakchoi can be promoted by adding biochar at a concentration of at least 3% (Table 2).

Table 2 Effects of different treatments on the growth of pakchoi.

Treatments	Height
(cm)	Root length
(cm)	Shoot dry weight (g plant−1 )	Root dry weight
(g plant−1 )	Chlorophyll a content
(mg g−1 FW)	Chlorophyll b content
(mg g−1 FW)	Total chlorophyll content
(mg g−1 FW)	
CK	16.54 ± 0.40 b	18.92 ± 2.36b	4.74 ± 0.11c	0.42 ± 0.04c	2.29 ± 0.21b	12.68 ± 0.88c	15.14 ± 1.17c	
T1	16.39 ± 0.50 b	23.41 ± 4.12a	5.23 ± 0.13bc	0.48 ± 0.01c	2.55 ± 0.06ab	13.23 ± 0.84bc	15.92 ± 0.72bc	
T2	18.06 ± 0.12 a	20.83 ± 1.60a	5.65 ± 0.21b	0.56 ± 0.03b	2.69 ± 0.12a	14.09 ± 0.35ab	16.79 ± 0.54ab	
T3	17.67 ± 0.29 a	23.91 ± 1.08a	7.25 ± 0.37a	0.73 ± 0.02a	2.46 ± 0.04ab	14.28 ± 0.23ab	16.61 ± 0.29bc	
T4	18.42 ± 0.74 a	22.82 ± 4.95a	6.63 ± 0.39a	0.70 ± 0.02a	2.65 ± 0.11a	15.46 ± 0.34a	18.21 ± 0.17a	
Notes.

Different lowercase letters on the same column indicate significant differences at a level of p < 0.05 (n = 3).

Effects of different cultivation modes on the quality of pakchoi

The experiments proved that the soluble sugar, soluble protein, VC, and cellulose contents in pakchoi leaves were significantly increased after biochar treatment compared with CK. The application of biochar at 3% (T2) and 7% (T4) treatments significantly increased the chlorophyll a, chlorophyll b, and total chlorophyll content in pakchoi leaves compared to the control without biochar application. Nitrate reductase activity in pakchoi leaves was significantly higher than in control and 1% biochar treatments when the amount of biochar added was more than 3%. With an increase in biochar content, nitrate content first decreased and then increased, and it was significantly reduced in the 3% biochar treatment. Therefore, the quality of pakchoi under the 3% biochar treatment was significantly better than that under the other treatments (Table 3).

Table 3 Effects of different cultivation modes on the quality of pakchoi.

Treatments	Soluble sugar (mg 100 g−1 )	Soluble protein (mg g−1 )	VC
(mg 100 g−1 )	Cellulose
(%)	Nitrate
(mg kg−1 )	Nitrite reductase
[µg g−1 h−1)]	
CK	25.76 ± 1.90b	1.44 ± 0.02b	4.32 ± 0.54b	1.70 ± 0.07d	286.93 ± 5.86b	50.79 ± 1.62b	
T1	41.01 ± 1.15a	1.55 ± 0.05b	3.06 ± 0.31b	2.58 ± 0.10c	257.71 ± 0.59c	57.54 ± 5.03b	
T2	39.18 ± 2.33a	1.65 ± 0.08a	5.50 ± 0.41a	3.96 ± 0.25b	225.67 ± 0.44d	75.89 ± 3.01a	
T3	14.46 ± 0.48d	1.56 ± 0.12b	6.30 ± 1.38a	4.56 ± 0.18a	301.48 ± 0.69a	88.86 ± 4.01a	
T4	20.01 ± 0.69c	1.54 ± 0.06b	5.40 ± 1.08a	2.89 ± 0.16c	288.08 ± 9.01b	86.84 ± 1.05a	
Notes.

Different lowercase letters on the same column indicate significant differences at a level of p < 0.05 (n = 3).

Effects of different concentrations of biochar on soil nutrient and enzyme activities

The urease, phosphatase, sucrase, catalase activity, and dehydrogenase of pakchoirhizosphere soil increased after adding biochar (Table 4). The dehydrogenase activity of pakchoi in the rhizosphere soil increased significantly with biochar application rate. Compared with the soil under control conditions, the urease, phosphatase, sucrase, and catalase activities in the soil first increased and then decreased with increased biochar added. These properties peaked in the soiltreated with 3% biochar, which was significantly higher than that of CK.

The pH value and contents of organic matter, alkali-hydrolyzable nitrogen, phosphorus, potassium, and EC values in the rhizosphere soil of pakchoi were enhanced after applying biochar (Table 5). Compared with the soil under control conditions, the contents of phosphorus, potassium, EC value, and organic matter in the rhizosphere soil of pakchoi gradually increased with the increase in biochar content. The highest values were found in the soil treated with 7% biochar and were significantly higher than in the control soil. The alkali-hydrolyzable nitrogen content and pH value were first enhanced and then reduced. Peak values were found in the soil under the 3% biochar treatment and were significantly higher than those without biochar addition.

Table 4 Application of different concentrations of biochar on soil enzyme activities.

Treatments	Urease activity
(NH3-Nmg g−1 soil d−1 )	Phosphatase activity
(mg phenol
g−1 soil d−1 )	Sucrase activity
(mg g−1 soil d−1 )	Catalase activity (0.05N KMnO4 ml g −1 soil)	Dehydrogenase
activity
(TPF mg g−1 soil)	
CK	1.06 ± 0.02b	2.13 ± 0.08 c	50.86 ± 4.89 d	36.16 ± 0.42 b	24.90 ± 1.62 d	
T1	1.09 ± 0.12ab	2.37 ± 0.03 b	98.53 ± 2.85 c	40.51 ± 2.86 b	33.45 ± 2.81 c	
T2	1.20 ± 0.00a	2.59 ± 0.11 a	122.80 ± 4.58 a	46.57 ± 2.88 a	39.37 ± 5.52 bc	
T3	1.21 ± 0.11 a	2.54 ± 0. 09 a	111.67 ± 0.94 b	47.59 ± 2.86 a	43.31 ± 2.13b	
T4	1.15 ± 0.06 ab	2.31 ± 0.07 b	114.07 ± 2.22 b	36.46 ± 2.48 b	52.88 ± 3.29 a	
Notes.

Different lowercase letters on the same column indicate significant differences at a level of p < 0.05 (n = 3).

Table 5 Application of different concentrations of biochar on soil nutrients.

Treatments	Available nitrogen (mg kg−1 )	Available phosphorus (mg kg−1 )	Available potassium
(mg kg−1 )	pH	EC
(µs cm−1 )	organic matter
(g kg−1 )	
CK	116.66 ± 0.88ab	68.14 ± 0.78c	196.23 ± 9.47c	8.21 ± 0.02 b	150.14 ± 6.78c	31.84 ± 2.02 c	
T1	119.85 ± 0.79a	68.12 ± 2.04c	216.30 ± 9.49bc	8.27 ± 0.02ab	146.12 ± 2.04c	35.51 ± 2.19 c	
T2	121.429 ± 1.36a	79.17 ± 1.35b	232.78 ± 9.53b	8.30 ± 0.01a	168.17 ± 1.35b	45.31 ± 2.21b	
T3	110.43 ± 5.37b	75.94 ± 2.87b	292.54 ± 16.81a	8.26 ± 0.01ab	175.94 ± 2.87b	53.32 ± 0.98 a	
T4	116.66 ± 2.70ab	88.12 ± 2.45a	312.75 ± 4.68a	8.30 ± 0.06a	188.12 ± 2.45a	57.06 ± 5.71 a	
Notes.

Different lowercase letters on the same column indicate significant differences at a level of p < 0.05 (n = 3).

Effects of different concentrations of biochar on soil bacterial community structure of pakchoi

After adding biochar, the structure of bacterial flora in the rhizosphere soil of pakchoi changed (Fig. 1). Among the relatively abundant dominant phyla Anaerolineae, the relative abundances of Gemmatimonadetes and Deltaproteobacteria were reduced, and Alphaproteobacteria, Gammaproteobacteria, Bacteroidia, Acidimicrobiia were increased. The microbiot structure changed significantly when the biochar content exceeded 3%.

Figure 1 Histogram of relative abundance of class-level species of bacteriaceae in the soil of pakchoi applied with different concentrations of biochar.

CK, biochar treatment was not applied; T1, treated with 1% biochar; T2, treated with 3% biochar; T3, treated with 5% biochar; T4, treated with 7% biochar. 1, 2, and 3 represent three replications.

Adding different biochar impacted the bacterial diversity in the rhizosphere soil of pakchoi significantly (Fig. 2), while adding 1% biochar (T1) had no significant change compared with the control, which was clustered together. The structure of the bacterial community in the rhizosphere soil of white pakchoi was significantly changed after adding 3% biochar (T2). The structure of the microbial community in the rhizosphere soil of white pakchoi treated with 3% biochar (T2), 5% biochar (T3), and 7% biochar (T4) was similar and clustered together. Microbial communities in the rhizosphere soil of pakchoi were classified into two groups. The first group was comprised Ilumatobacter, Luteolibacter, Lysobacter, Arthrobacter, Mesorhizobium, Pontibacter, Lactobacillus, Faecalibacterium, and Bifidobacterium, and the relative abundance of these bacteria increased significantly. The second group comprises the sweet and sour bacteria Gemmatimonas, Bacillus, Stenotrophobacter, Ramlibacter, Bryobacter, Actinoplanes, Comamonas, Flavisoib, Megamonas, Steroldobacter, Ohtaekwangia, and Luteitalon. These were clustered together, and the relative abundance of these bacteria was significantly reduced. Adding 3% biochar significantly changed the microbial community structure in the rhizosphere soil of pakchoi plants.

Figure 2 Class level species abundance clustering heatmaps of different concentrations of biochar on soil bacterium taxa distribution of pakchoi.

CK, biochar treatment was not applied, CK1, CK2, and CK3 represent three replications; T1, treated with 1% biochar, T11, T12, and T13 represent three replications; T2, treated with 3% biochar, T21, T22, and T23 represent three replications; T3, treated with 5% biochar, T31, T32, and T133 represent three replications; T4, treated with 7% biochar, T41, T42, and T43 represent three replications.

Cluster analysis depicted that 1% biochar (T1) clustered together with the control group withsimilar bacterial community structures. A biochar content of >3% (T2) was also clustered. However, the structure of the bacterial community differed from that of the control group. All treatments were grouped into two types: (i) the control (CK) and 1% (T1) biochar, and (ii) the treatment above 3% biochar (T2, T3, and T4), among which the abundance of Gemmatimonas was significantly reduced after treatment with more than 3% biochar (T2) (Fig. 3). Analysis of similarities (ANOSIM) compared the difference in beta diversity between samples from different groups. The boxplot results revealed significant differences between the T2 and T3 treatment groups and the control group (Fig. 4A). Principal component analysis demonstrated that CK and T1 significantly differed from T2, T3, and T4 in the PC1 axis when they were clustered together three times. Adding 1% biochar had no significant effect on the structure of the soil microbial community, and adding more than 3% led to significant changes in the structure of the soil microbial community (Fig. 4B).

Figure 3 Bacterial abundance and cluster analysis of Pakchoi soil with different concentrations of biochar (CK, TI, T2, T3, and T4).

CK, biochar treatment was not applied; T1, treated with 1% biochar; T2, treated with 3% biochar; T3, treated with 5% biochar; T4, treated with 7% biochar.

Figure 4 (A) Anosim analysis box diagram and. (B) principal component analysis (PCA) plot of in Rhizosphere soil of pakchoi under different treatments (CK, T1, T2, T3, and T4).

CK, biochar treatment was not applied; T1, treated with 1% biochar; T2, treated with 3% biochar; T3, treated with 5% biochar; T4, treated with 7% biochar.

There was a significant positive correlation between Gemmatimonas and nitrate content and a significant negative correlation between Gemmatimonas and pH, NR, SDW, and RDW. There was a significant negative correlation between Ilumatobacter and nitrate content and a significant positive correlation between Ilumatobacter and pH, NR, SDW, and RDW. The control, without biochar (CK), was positively correlated with nitrate and VC. Biochar treatment (T2) was positively correlated with soil pH and NR, and biochar treatments (T3 and T4) were also positively correlated with COSS, nr, and urease activities. Therefore, biochar application is closely related to dry weight, nitrate accumulation, and root DW. In addition, COSS, NR, pH, and urease activity were the main influencing factors (Fig. 5).

Figure 5 Redundancy analysis (RDA) of the relationship of the pakchoi quality with soil properties and (A) soil-specific microbial species and (B) different treatment of biochar.

Soil properties included pH, organic matter (OM), shoot dry weight (SDW), root dry weight (RDW), nitrite reductase (NR), the content of soluble sugar (COSS), Vitamin C (VC), and nitrate of bacterial community changes with soil parameters (CK, T1, T2, T3, and T4).

The control rhizosphere soil had significantly higher carbohydrate transport and metabolism, amino acid transport and metabolism, general function prediction, energy production and conversion, post-translational modification, protein turnover, and chaperones than T2 treatment. The T2 rhizosphere soil had significantly higher lipid transport and metabolism, transcription, cell motility, signal transduction metabolism, intracellular trafficking, secretion, and vesicular transport than CK (Fig. 6).

Figure 6 Differential microorganisms KEGG annotation classification (CK1 vs. T21).

CK1, biochar treatment was not applied; T21, treatment with 3% biochar.

The control rhizosphere soil had significantly higher carbohydrate transport and metabolism,amino acid transport and metabolism, and general function predictions than T3 treatment. The T3 rhizosphere soil had significantly higher intracellular trafficking, secretion, vesicular transport, secondary metabolite biosynthesis, transport and catabolism, cell motility, lipid transport and metabolism, and transcription than the control (CK) (Fig. 7).

Figure 7 Differential microorganisms KEGG annotation classification (CK1 vs. T31).

CK1, biochar treatment was not applied; T31, treatment with 5% biochar.

Carbohydrate metabolism, membrane transport, cellular community-prokaryotes, energymetabolism, biosynthesis of other secondary metabolites, xenobiotic biodegradation and metabolism, lipid metabolism, and global and overview map control (CK) were significantly higher than those in treatment T4 (7%). The following metabolic processes were significantly higher in T4 (7%)-treated rhizosphere soil than in control soil (CK): nucleotide metabolism, glycan biosynthesis and metabolism, cell motility, replication and repair, cell growth and death, signal transduction, amino acid metabolism, metabolism of cofactors and vitamins, metabolism of other amino acids, and translation (Fig. 8). Therefore, carbohydrate transport and metabolism in the rhizosphere soil of pakchoi decreased, and lipid transport and metabolism increased after applying biochar. These metabolic changes may explain the increase in soil nutrients after applying biochar.

Figure 8 Differential microorganisms KEGG annotation classification (CK1 vs. T41).

CK1, biochar treatment was not applied; T41, treatment with 7% biochar.

Discussion

Biochar promotes the growth of pakchoi and improves its quality

Several studies have focused on the effect of biochar on vegetable plant growth, yield, andquality. A study demonstrated that the combination of 35 t ha−1 biochar and 200 kg ha−1 N fertilizers significantly increased tomato yield, VC content, sugar-acid ratio, and economic benefits (Guo et al., 2021). Wood chip biochar enhanced the growth of ice plants in coastal soil. Adding wood chip biochar improved the nutritional quality of the plants, regardless of whether chemical fertilizer was applied (You et al., 2021). The results of this study confirmed the hypothesis that the application of 3% biochar enhanced the growth and chlorophyll content of pakchoi, improved the nutritional quality of pakchoi (Table 3). The positive effect of biochar amendment on pakchoi growth and quality may be attributed to improvements in soil properties (Zheng et al., 2018), improved soil bulk density and water retention (Wang et al., 2022), enhanced nutrient availability (Amin, 2020), alterations in the soil microbial community (Lu et al., 2020). Research has shown that biochar can promote the development of plant roots, and then enhance the photosynthesis of leaves, so that plants can grow healthily, which is conducive to the plant production and the cultivation of soil (Ren et al., 2021). The results of this study show that indicate that 3% biochar significantly increased root length and chlorophyll content of pakchoi (Tables 2 and 3), this may be one of the main reasons for promoting the growth and improving the quality of pakchoi.

Biochar enhances soil nutrient availability and reduces nitrate in pakchoi

Biochar has a loose and porous structure, and its addition to the soil can improve the soil structure and enhance soil fertility (Gorovtsov et al., 2020), fix nutrients in the soil, and increase the content of available nutrients (Zhang et al., 2019). The high carbon content of biochar can replenish the organic carbon in the soil and increase the soil organic matter content; biochar applied at certain concentrations gradually absorbed the nitrogen released from the N, P, and K compound fertilizers, reducing the nitrogen amount absorbed by plants and decreasing the nitrate content in plants (Yousaf et al., 2016). Because biochar is alkaline, its application to the soil can increase soil pH (Liu, Liu & Zhang, 2014). Moreover, the relatively large surface area of biochar addition based on biochar in the short term can improve soil quality by neutralizing soil pH, increasing soil nutrient contents (Kong et al., 2021), and promoting plant growth. In this study, under weakly alkaline soil conditions, adding 3% biochar to the rhizosphere soil of pakchoi resulted in a peak alkaline nitrogen content, a significant increase in soil pH (Tables 3 and 5), and nutrient balance, which was beneficial for the growth and quality improvement of pakchoi. Biochar can significantly promote the growth of plant roots (Table 2), and an increase in root biomass can increase the secretion of secretions (Wang et al., 2018), thereby altering the abundance of rhizosphere soil microorganisms and increasing soil enzyme activity (Feng et al., 2021). In this study, 3% biochar significantly increased soil urease activity (Table 4) and nitrogen availability in the soil (Table 5). Studies have illustrated the combined action of biochar and nitrogen-fixing bacteria on microbial and enzymatic activities of soil N cycling (Gou et al., 2022). Application of biochar accelerates the metabolism of soil biota, turning more nitrogen from fertilizers into organic forms; hence, less mineral nitrogen is left for plant intake, which reduces nitrate levels in red beet (Maroušek et al., 2018). It can increase the utilization rate of nitrogen in the soil and reduce nitrate accumulation in the leaves of pakchoi (Table 3). However, the improved soil nutrient metabolism and microbial community structure caused by adding biochar require further in-depth research.

Correlation analysis between biochar changes in soil nutrient metabolism and microbial community structure

Biochar amendments can alter plant rhizosphere microbiome, which profoundly affects plant growth and fitness (Jin et al., 2023). Regulation of C/N ratio by adding biochar and N fertilizer affects the composition and diversity of soil bacterial communities (Kong et al., 2021). This study demonstrated that the relative abundance of the bacterial community in pakchoi rhizosphere soil was changed by applying more than 3% biochar. Among the relatively abundant dominant phyla, Anaerolineae, Gemmatimonadetes, Deltaproteobacteria, and Verrucomicrobiae were reduced, and 303 Alphaproteobacteria, Gammaproteobacteria, Bacteroidia, and Acidimicrobiia relative abundance increased (Fig. 1). Biochar affects the symbiotic pattern of microorganisms, increasing the proportion of positive interactions in the microbial community (He et al., 2021), the advantages of these changes will be verified in the future.

Previous research has revealed that biochar amendment significantly increases the relativeabundance of the potential biocontrol bacteria Bacillus and Lysobacter (Feng et al., 2021). Bacteria, including species within the genera Arthrobacter and Bacillus, can solubilize phosphate or mitigate salinity stress (Jiang et al., 2019; Tchakounté et al., 2020). The abundance of some potentially beneficial bacteria, such as Luteolibacter, Glycomyces, Flavobacterium, and Flavihumibacter, increases in organic fertilizer-amended soil (Huang et al., 2022a; Huang et al., 2022b). The application of both types of biochar increases the nodule number by 52% under well-watered and drought conditions by improving the symbiotic performance of chickpeas with Mesorhizobium ciceri (Egamberdieva et al., 2019). Research displays that Gemmatimonas possess unique characteristics for assimilative and dissimilative N processes, with new implications for cultivation strategies to better assess the metabolic abilities of Gemmatimonadetes (Chee-Sanford, Tian & Sanford, 2019). In addition, the relative abundance of Gemmatimonas was reduced, whereas the relative abundances of Ilumatobacter, Luteolibacter, Lysobacter, Arthrobacter, Mesorhizobium, and other bacteria in soil supplemented with 3% biochar were significantly increased (Fig. 2). The increase in these five (Ilumatobacter, Luteolibacter, Lysobacter, Arthrobacter, and Mesorhizobium) beneficial bacteria after adding biochar might be related to increased soil available nutrients and improved pakchoi quality. Gemmatimonas species carry genes that promote mineralization, nitrification, dissimilatory/assimilatory nitrate reduction, denitrification, anammox reactions, and N fixation. The functional microbes Gemmatimonas reduce various N conversion processes at different rates in biochar-added soil, which might reduce nitrate accumulation in pakchoi (Zhou et al., 2022a; Zhou et al., 2022b). The bacterial diversity was significantly improved, growth-promoting bacteria, such as Gemmatimonadetes and Proteobacteria, became more abundant, and Acidobacteria and Actinobacteria became less abundant after adding biochar (Feng et al., 2021). Xu et al. (2016) demonstrated that under biochar application conditions, the relative abundance of Acidobacteria, Chloroflexi and Gemmatimonadetes decreased, while Proteobacteria, Bacteroidetes and Actinobacteri a increased. The results were not nearly identical and may be related to the soil type and original flora of the test site.

Adding modified biochar to the soil changed the relative abundance of dominant bacterialcommunities and increased the relative abundance of some functional bacterial communities (Hua et al., 2021). In this study, the nitrate content was positively correlated with the abundance of Gemmatimonadetes (Fig. 5A), and the reduction in nitrate accumulation by biochar addition was closely related to the decrease in Gemmatimonadetes abundance. Biochar enhances the retention capacity of nitrogen fertilizers and affects the diversity of nitrifying functional microbial communities in soil (Zhang et al., 2021). However, the role of microorganisms in soil metabolism requires further verification. Ilumatobacter is a new genus of acid microorganisms that have been isolated from the sea by scholars (Matsumoto et al., 2013). In this study, the nitrate content was significantly negatively correlated with the relative abundance of Ilumatobacte and positively correlated with aboveground dry matter accumulation and soluble sugar content (Fig. 5A). This indicates that the increase in the abundance of this genus is closely related to the growth and quality improvement of pakchoi.

Conclusion

Our results provide experimental evidence that biochar application In the production of pakchoi can improve soil available nutrient content and soil enzyme activity, the application biochar affected the abundance of dominant bacterial groups (e.g., Gemmatimonadetes and Ilumatobacter) communities in the soil.The alterations in soil microbial metabolic pathways results in an enhanced soil nutrient environment, thereby facilitating nutrient uptake by crops, augmenting crop productivity, mitigating nitrate accumulation in leaves, and significantly enhancing of pakchoi, the effects of incorporating 3% biochar are particularly pronounced.

Supplemental Information

Supplemental Information 1 The experimental process and results

Supplemental Information 2 The first raw data obtained from high-throughput sequencing of the first repeated soil sample compared in the experiment

CK, biochar treatment was not applied; 1, 2, and 3 represent three replications.

Supplemental Information 3 The second raw data obtained from the first repeated soil sample high-throughput sample compared in the experiment

CK, biochar treatment was not applied; 1, 2, and 3 represent three replications.

Supplemental Information 4 The first raw data obtained from the second repeated soil sample high-throughput sequencing of the control in the experiment

Supplemental Information 5 The second raw data obtained from the second repeated soil sample high-throughput sequencing of the control in the experiment

CK, biochar treatment was not applied; 1, 2, and 3 represent three replications.

Supplemental Information 6 The first raw data obtained from the third repeated soil sample high-throughput sequencing of the control in the experiment

CK, biochar treatment was not applied; 1, 2, and 3 represent three replications.

Supplemental Information 7 The second raw data obtained from the third repeated soil sample high-throughput sequencing of the control in the experiment

CK, biochar treatment was not applied; 1, 2, and 3 represent three replications.

Supplemental Information 8 The first raw data obtained from the first repeated soil sample high-throughput sequencing of T1 treatment in the experiment

T1, treated with 1% biochar. 1, 2, and 3 represent three replications.

Supplemental Information 9 The second raw data obtained from the first repeated soil sample high-throughput sequencing of T1 treatment in the experiment

T1, treated with 1% biochar. 1, 2, and 3 represent three replications.

Supplemental Information 10 The first raw data obtained from the second repeated soil sample high-throughput sequencing of T1 treatment in the experiment

T1, treated with 1% biochar. 1, 2, and 3 represent three replications.

Supplemental Information 11 The second raw data obtained from the second repeated soil sample high-throughput sequencing of T1 treatment in the experiment

T1, treated with 1% biochar. 1, 2, and 3 represent three replications.

Supplemental Information 12 The first raw data obtained from the third repeated soil sample high-throughput sequencing of T1 treatment in the experiment

T1, treated with 1% biochar. 1, 2, and 3 represent three replications.

Supplemental Information 13 The second raw data obtained from the third repeated soil sample high-throughput sequencing of T1 treatment in the experiment

T1, treated with 1% biochar. 1, 2, and 3 represent three replications.

Supplemental Information 14 The first raw data obtained from the first repeated soil sample high-throughput sequencing of T2 treatment in the experiment

T2, treated with 3% biochar. 1, 2, and 3 represent three replications.

Supplemental Information 15 The second raw data obtained from the first repeated soil sample high-throughput sequencing of T2 treatment in the experiment

T2, treated with 3% biochar. 1, 2, and 3 represent three replications.

Supplemental Information 16 The first raw data obtained from the second repeated soil sample high-throughput sequencing of T2 treatment in the experiment

T2, treated with 1% biochar. 1, 2, and 3 represent three replications.

Supplemental Information 17 The second raw data obtained from the second repeated soil sample high-throughput sequencing of T2 treatment in the experiment

T2, treated with 3% biochar. 1, 2, and 3 represent three replications.

Supplemental Information 18 The first raw data obtained from the third repeated soil sample high-throughput sequencing of T2 treatment in the experiment

T2, treated with 3% biochar. 1, 2, and 3 represent three replications.

Supplemental Information 19 The second raw data obtained from the third repeated soil sample high-throughput sequencing of T2 treatment in the experiment

T2, treated with 3% biochar. 1, 2, and 3 represent three replications.

Supplemental Information 20 The first raw data obtained from the first repeated soil sample high-throughput sequencing of T3 treatment in the experiment

T3, treated with 5% biochar. 1, 2, and 3 represent three replications.

Supplemental Information 21 The second raw data obtained from the first repeated soil sample high-throughput sequencing of T3 treatment in the experiment

T3, treated with 5% biochar. 1, 2, and 3 represent three replications.

Supplemental Information 22 The first raw data obtained from the second repeated soil sample high-throughput sequencing of T3 treatment in the experiment

T3, treated with 5% biochar. 1, 2, and 3 represent three replications.

Supplemental Information 23 The second raw data obtained from the second repeated soil sample high-throughput sequencing of T3 treatment in the experiment

T3, treated with 5% biochar. 1, 2, and 3 represent three replications.

Supplemental Information 24 The first raw data obtained from the third repeated soil sample high-throughput sequencing of T3 treatment in the experiment

T3, treated with 5% biochar. 1, 2, and 3 represent three replications.

Supplemental Information 25 The second raw data obtained from the third repeated soil sample high-throughput sequencing of T3 treatment in the experiment

T3, treated with 5% biochar. 1, 2, and 3 represent three replications.

Supplemental Information 26 The first raw data obtained from the first repeated soil sample high-throughput sequencing of T4 treatment in the experiment

T4, treated with 7% biochar. 1, 2, and 3 represent three replications.

Supplemental Information 27 The second raw data obtained from the first repeated soil sample high-throughput sequencing of T4 treatment in the experiment

T4, treated with 7% biochar. 1, 2, and 3 represent three replications.

Supplemental Information 28 The first raw data obtained from the second repeated soil sample high-throughput sequencing of T4 treatment in the experiment

T4, treated with 7% biochar. 1, 2, and 3 represent three replications.

Supplemental Information 29 The second raw data obtained from the second repeated soil sample high-throughput sequencing of T4 treatment in the experiment

T4, treated with 7% biochar. 1, 2, and 3 represent three replications.

Supplemental Information 30 The first raw data obtained from the third repeated soil sample high-throughput sequencing of T4 treatment in the experiment

T4, treated with 7% biochar. 1, 2, and 3 represent three replications.

Supplemental Information 31 The second raw data obtained from the third repeated soil sample high-throughput sequencing of T4 treatment in the experiment

T4, treated with 7% biochar. 1, 2, and 3 represent three replications.

Additional Information and Declarations

Competing Interests

Author Contributions

Data Availability

The authors declare there are no competing interests.

Xia Wu conceived and designed the experiments, performed the experiments, analyzed the data, prepared figures and/or tables, national Natural Science Foundation of China (funding code: 31801905);Natural Science Foundation of Heilongjiang Province, China (funding code: LH2021C066), and approved the final draft.

Fengjun Yang conceived and designed the experiments, prepared figures and/or tables, authored or reviewed drafts of the article, and approved the final draft.

Jili Zhang analyzed the data, prepared figures and/or tables, authored or reviewed drafts of the article, and approved the final draft.

Feng Gao performed the experiments, analyzed the data, prepared figures and/or tables, and approved the final draft.

Yi chen Hu performed the experiments, analyzed the data, prepared figures and/or tables, and approved the final draft.

Kejun Yang conceived and designed the experiments, authored or reviewed drafts of the article, and approved the final draft.

Peng Wang conceived and designed the experiments, authored or reviewed drafts of the article, natural Science Foundation of Heilongjiang Province, China (funding code: LH2021C066), and approved the final draft.

The following information was supplied regarding data availability:

The data is available at NCBI SRA: PRJNA962023.

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
