# Peer review of "Biochar’s role in improving pakchoi quality and microbial community structure in rhizosphere soil"

_PeerJ, doi:10.7717/peerj.16733_

## Round 0.1 · original submission · Major Revisions

Dear Dr. Wu,

Thank you for your submission to PeerJ.

Your paper has been evaluated by experts in the field. According to the reviewers' report, it requires a number of Major Revisions.

Reviewer 1 ·

Basic reporting

Comment 1: Should be add one paragraph about effect of biochar on physiological traits in plants. Please include the following recent studies in the literature (https://doi.org/10.3390/agronomy11112341; https://doi.org/10.3390/horticulturae7080250; doi: 10.3389/fpls.2022.947547 ).
Comment 2: The significance of the study and a solid hypothesis of the present study at the end of the introduction should be provided to give the reader more information regarding the purpose of the study and the mechanistic used to achieve this objective.

Experimental design

Commen 3: In methodology part, authors should be provided biochar properties.
Comment 4: Why 4 levels of biochar (1, 3, 5 and 7%) were used in the experiment?
Comment 5: In methodology part, authors should be provided full method about soluble sugar and nitrate contents in the leaves.
Comment 6 : Authors should be provided full method about the soluble protein content and the cellulose content.
Comment 7: In methodology part, the physiological traits were evaluated in what growth stage?

Validity of the findings

Results and Analysis
Comments 8: In Results and Analysis part, physiological properties such as chlorophyll a, chlorophyll b, and total chlorophyll contents are missing.
Discussion
Comment 9: In the discussion section, is needed an explanation about how biochar affect the plant growth, physiological and biochemical properties in plants.
Comment 10: What are the mechanisms that allow biochar to increase plant growth and soil enzyme activities?
Comment 11: The conclusion needs to be rewritten.

Annotated reviews are not available for download in order to protect the identity of reviewers who chose to remain anonymous.

Reviewer 2 ·

Basic reporting

The English in the manuscript is good. The background and significance of the study have been explained well. The manuscript is written well.

Experimental design

The experimental part is well written. Some suggestions have been provided for the experimental part.

Validity of the findings

The findings of the current work are exciting and will be helpful in the field of agricultural research in order to improve the yield of vegetables and grains.

Additional comments

Comments on manuscript:
1 Check scientific name of cabbage in lines 11-12 and throughout the manuscript.
2 Why rhizosphere soil microorganisms are limited in alkaline soil?
3 What was the rationale for choosing biochar in the current study?
4 What does VC mean in line number 19?
5 Rewrite conclusion in concrete statement from line 32-36.
6 Look for spacing issue in line 49,50.
7 Line 116 authors used 105°C for 30 minutes to remove water from plant but I think may be plant phytochemical may get degraded at this temp? Is it random selection of temperature or authors have taken idea from somewhere else?
8 Spacing issue between line 121 and 122.
9 Which method was used to isolate genomic DNA? Line 143,144
10 What was the relevance of using its1737f - its2-2043r primer in this study? Are they bacterial primer or fungal primer? Line 147.
11 Mention PCR conditions in line 147.
12 Space between line 266, 267

Annotated reviews are not available for download in order to protect the identity of reviewers who chose to remain anonymous.

·

Basic reporting

In this study, the authors have evaluated the effects of different concentrations of bio char on growth and productivity of Chinese cabbage. Simultaneously, a correlation has been tried to establish among different biochemical and microbiological properties of cabbage cultivated soil and the optimum bio char concentration. The addition of bio char into soil for different benefactions is a well-established technology; hence this manuscript lacks a serious and innovative novelty in this regard. Though, the evaluation of bio char concentration for cabbage cultivation could be a significant step for successful propagation of this technology into different aspects of agricultural studies.

Experimental design

The experiments are well framed but, methodology needs to be precisely expanded to widen the ambit of this study. The results aren’t described thoughtfully, and there are some pernicious ambiguities also in the description and inferences of out comings.

Validity of the findings

The experiments are well framed but, methodology needs to be precisely expanded to widen the ambit of this study. The results aren’t described thoughtfully, and there are some pernicious ambiguities also in the description and inferences of out comings.

---

## Round 0.2 · accepted · Accept

I confirm that the authors have revised the manuscript as per the comments or suggestions of the Reviewers